# Detection of Moving Load on Pavement Using Piezoelectric Sensors

**DOI:** 10.3390/s20082366

**Published:** 2020-04-22

**Authors:** Tao Xiang, Kangxu Huang, He Zhang, Yangyang Zhang, Yinnan Zhang, Yuhui Zhou

**Affiliations:** 1College of Civil Engineering and Architecture, Zhejiang University, Hangzhou 310058, China; xiangtao@zju.edu.cn (T.X.); hkx57@zju.edu.cn (K.H.); zhangyinnan@zju.edu.cn (Y.Z.); 11812062@zju.edu.cn (Y.Z.); 2Faculty of Mechanical Engineer and Mechanics, Ningbo University, Ningbo 315211, China; zhangyangyang@nbu.edu.cn

**Keywords:** PZT sensor, load identification, weigh-in-motion

## Abstract

More and more researches have been carried out recently on Weigh-In-Motion (WIM) technology for solving the traffic safety problems caused by overload. In this article, we aim to study the measurement accuracy of the WIM system. Based on the electromechanical theory and elastic half-space method, we establish a theoretical model of multi-layer structure to investigate the correlation between the output voltage of the piezoelectric sensor and the applied force. In addition, we performed cyclic and moving load experiments to verify the accuracy of the analytical calculations. The load magnitude identified by this theoretical model matched the experiments very well, which shows that this model is effective for the WIM system. In addition, we proved that the load frequency is an important factor affecting the measurement accuracy of the sensor, which further enables us to design more suitable sensors for certain use scenarios.

## 1. Introduction

Overloading is one of the main factors of traffic accidents, causing severe economic losses and casualties, and also significantly reducing the service life of roads and bridges. To ensure driving safety, the static weighing scale has been widely used at toll station on highways. The vehicle gross weight can be measured with high accuracy in this way, but it cannot get the single axle weight. Moreover, there are several obvious disadvantages including low efficiency, high expense and even causing traffic jams. During the past decades, the traffic flow has been increasing continuously, which makes the static weighing methods no longer able to meet the requirements in some scenarios.

As an improvement, the weigh-in-motion (WIM) method [1] may directly measure the wheel load when the vehicles are in motion [2]. This makes the WIM method superior to the static weighing method with time-saving and higher traffic efficiency. Given the above advantages, the WIM system has been widely studied and developed in recent years [3,4,5,6,7,8]. However, the WIM method encounters various shortcomings, e.g., high costs for installation and maintenance, and low accuracy of measurements [2]. Among the numerous factors affecting accuracy, there are two main factors: the reliability of sensors and corresponding algorithms [7]. With the development of computer technology, there are many researches about the algorithms conducted for the sake of improving the accuracy of WIM systems. For example, when applied identifying the axle weight of moving vehicles on bridges, Zhao et al. [9] proposed an improved algorithm for the commercial bridge weigh-in-motion (B-WIM). Kim et al. [10] introduced the artificial neural networks (ANN) into the B-WIM for long-span bridges and more accurate results were obtained. Jia et al. [11] contributed to further improvements on the B-WIM by involving the back-propagation neural network (BPNN) with a recognition error of less than 5%. Zhao et al. [12] proposed an algorithm based on a modified 2-D Moses algorithm to identify the vehicular axle weights and the identified results of the filed test showed agreement with the static method. Xia et al. [13] present a traffic load identification methodology using multiple strain sensors and computer vision technology that can implement traffic information collection and load monitoring in complex traffic scenarios. González et al. [14] performed an evaluation of an ANN technique applied to multiple-sensor WIM systems and the results demonstrated that the algorithm was potential in improving WIM accuracy. In fact, the load identification problem of WIM system can be classified as an inverse problem [15,16]. These recognition models mentioned in the previous researches normally are trained by already existing data. Hence, the accuracy performance of the recognition models heavily relied on training data. Meanwhile, the data obtained from experiments or numerical models is inefficient because it is time consuming [17].

There are mainly three types of sensors in the aforementioned or other researches used in WIM systems: strain sensor [8,13], piezoelectric sensor [4], and quartz sensor [18]. Among the available sensors, piezoelectric sensors have been attracting increasing interests due to their ability of self-powering and fast response to impact loadings [19,20,21,22,23], and, hence, have been widely applied in health monitoring systems for infrastructures [24,25,26,27]. Tang et al. [28] used integrated Lead Zirconium Titanate (PZT) sensors for comprehensive structural health monitoring. Hou et al. [29] proposed a smart aggregate based on PZT that can monitor the compressive seismic stress for low- and middle-rise architectural structures under moderate earthquakes. Liang et al. [30] utilized the piezoceramic transducers and time-reversal technique to monitor the load of the pin-connected structure. Yan et al. [31] proposed a structural health monitoring system based on piezoelectric ceramics for concrete crack detection. Divsholi [32] using PZT transducers and combining the wave transmission technique and the wave propagation technique developed a way to assess the local and overall conditions of a concrete beam.

There are mainly three types of commercial WIM equipment at present: curved plate, scale table and piezoelectric quartz. It is difficult to be widely used due to the expensive price. Therefore, there is an urgent need to resolve the contradiction between the price and accuracy of WIM system. In this paper, we proposed a model based on the electromechanical model and multilayer elastic half-space theory which can correlate the output voltage signals generated by the PZT sensor with the stress applied to it. Then, entering the voltage signals induced by a series of laboratory test experiments, the calculated results showed that high accuracy of identification can be obtained by this model.

## 2. Experiments

In laboratory tests, cyclic loads and moving loads experiments were performed to verify the accuracy of the identification theory. The cyclic loads experiment included the wheel tracking test and Mechanical Testing Simulation (MTS) test (Figure 1a). The differences between the wheel tracking test and the MTS test are the loading frequency and area. The actual vehicle speed and different axial load effects can be simulated by changing the frequency and the load of the MTS test machine respectively. Figure 1b describes the distribution of piezoelectric sensors that are encapsulated between two MC nylon plates, where the diameter of the circle equals the MTS loading punch. The wheel tracking test was performed according to the previous research work on energy harvesting [33] from the same group, and, hence, the details of the test are not repeated here for brevity.

Two different piezoelectric sensors including PZT chip and PZT bimorph were, respectively, used for cyclic loads and moving loads experiments. The positive and negative electrodes of each PZT sensors were connected by wires, which can output electrical signals to the oscilloscope when the PZT sensors were subjected to an external force. To investigate the effect of embedded depth of PZT sensors on the results, the PZT chips were embedded at four different depths (10 mm, 20 mm, 30 mm, 40 mm). The geometric and electrical parameters of piezoelectric sensors were described in Table 1.

Compared with cyclic loads experiments, the moving loads experiments revealed the effect of the multi-axis effect on the actual road surface. Figure 2 describes a load trolley moving on a steel rail embedded with a PZT bimorph and the voltage signals were recorded by a computer. The PZT bimorphs were embedded in the steel rail at depth 2 mm along the track and covered by cement. The trolley had two axles moving uniformly along the track under the traction of the rope. The voltage signals were collected and transmitted to the computer by a microcontroller unit (MCU) when the trolley passed by the PZT bimorph. Through the processing of voltage signals, the weight of the trolley and the overloaded warning information can be shown on the computer screen.

## 3. Theoretical Formulations

### 3.1. Piezoelectric Theory

The identification of cyclic loads and moving loads using piezoelectric sensors will be demonstrated based on the general theory of piezoelectricity [34] and the following assumptions:

(1) The PZT’s electromechanical responses do not affect the mechanical behavior of the tracking board;

(2) The dynamic effects of the system are ignored and the electromechanical procedure of the system is in a quasi-static state;

(3) The PZT chip has the same strain with rutting plate at the same place, that is $\varepsilon_{p}=\varepsilon_{b}$, where $\varepsilon_{p}$ and $\varepsilon_{b}$ are, respectively the column matrices strain of PZT sensor and rutting plate.

(4) The contact length (~20 mm) between the wheel and the rutting plate is comparable with the diameter of PZT chip, the PZT chip is therefore assumed in a state of simple compression through thickness direction.

The constitutive equations for the piezoelectric sensors in three-dimensional deformation [34] are described in Equation (Equation 1).
(1)$$\begin{array}{c} \sigma =c\varepsilon -dE\\ D=d\sigma +\mu E \end{array}$$
where the ε, σ, D and E are, respectively, column matrices of strain, stress, electric displacement, and electric intensity, and c, d, and μ are, respectively, the matrix of compliance coefficient, piezoelectric constant and dielectric constant at a fixed stress. The thickness-wise electric displacement can be read as
(2)$$D_{z}=\bar{d}\varepsilon +\bar{\mu }E_{z}$$
where $\bar{d}=dc$ and $\bar{\mu }=\mu -d^{2}$ are respectively the equivalent piezoelectric and dielectric constant. $E_{z}=V/h_{p}$, is the electric field intensity, where $h_{p}$ is the sensor thickness and *V* is the voltage drop across the sensor thickness. According to the theory of piezoelectricity, the electric charge generated at the electrode-covered surface of the PZT sensor is $Q=D_{z}A_{p}$, where $A_{p}$ is the effective area of PZT sensor covered by the electrode. When the PZT sensor is connected to an oscilloscope, the current generated in the circuit may be defined as I=-dQ/dt=V/R, where *R* is the electrical resistive load of the oscilloscope. Combining the expression of current and strain correlation with Equation (Equation 2), yields the following differential equation about the output voltage *V* from PZT sensor
(3)$$\frac{dV}{dt}+\frac{h_{p}}{\bar{\mu }RA_{p}}V=-\frac{\bar{d}h_{p}}{\bar{\mu }}\frac{d\varepsilon }{dt}$$

### 3.2. Identification Theory of Tracking Board Test

Based on previous assumptions, the PZT sensor is in a state of simple compression. Considering the Poisson effect (ν = 0.3) in the wheel tracking test, the thickness-wise electric displacement may be read as
(4)$$D_{z}={\bar{d}}_{31}\left(\varepsilon_{x,p}+\varepsilon_{y,p}\right)+{\bar{d}}_{33}\varepsilon_{z,p}+{\bar{\mu }}_{33}\frac{V}{h_{p}}$$
where ${\bar{d}}_{31}=d_{31}c_{11}+d_{31}c_{12}+d_{33}c_{13}$, ${\bar{d}}_{33}=2d_{31}c_{13}+d_{33}c_{13}$, ${\bar{\mu }}_{33}=\mu_{33}-d_{33}d_{31}-2d_{31}^{2}$, $\varepsilon_{x,p}=\varepsilon_{y,p}=\nu \varepsilon_{z,p}$. Correspondingly, the differential equations about voltage and strain can be read as
(5)$$\frac{dV}{dt}+\frac{Vh_{p}}{{\bar{\mu }}_{33}RA_{p}}=-\frac{h_{p}{\bar{d}}_{31}}{{\bar{\mu }}_{33}}\frac{d\varepsilon_{x,p}}{dt}-\frac{h_{p}{\bar{d}}_{31}}{{\bar{\mu }}_{33}}\frac{d\varepsilon_{y,p}}{dt}-\frac{h_{p}{\bar{d}}_{33}}{{\bar{\mu }}_{33}}\frac{d\varepsilon_{z,p}}{dt}$$

According to the previous assumptions, the strain of PZT sensor ($\varepsilon_{p}$) is the same with the rutting plate ($\varepsilon_{b}$) at the same place. In the cylindrical coordinate system, according to Hooke’s law, the strain of rutting plate can be read as
(6)$$\begin{array}{c} \varepsilon_{z,b}\left(t\right)=\frac{1}{E}\left[\sigma_{z}\left(t\right)-\nu \left(\sigma_{r}\left(t\right)+\sigma_{\theta }\left(t\right)\right)\right]\\ \varepsilon_{r,b}\left(t\right)=\frac{1}{E}\left[\sigma_{r}\left(t\right)-\nu \left(\sigma_{z}\left(t\right)+\sigma_{\theta }\left(t\right)\right)\right] \end{array}$$
where $\sigma_{z}\left(t\right)$, $\sigma_{r}\left(t\right)$, $\sigma_{\theta }\left(t\right)$ and *E* are, respectively, the stress in cylindrical coordinate system, Young’s modulus of rutting plate. Converting strain from cylindrical coordinate system to cartesian coordinate system, the strain of PZT sensor can be read as
(7)$$\begin{array}{c} \varepsilon_{z,p}\left(t\right)=\varepsilon_{z,b}\left(t\right)\\ \varepsilon_{x,p}\left(t\right)=\varepsilon_{y,p}\left(t\right)=\frac{1}{\sqrt{2}}\varepsilon_{r,b}\left(t\right) \end{array}$$

In the wheel tracking test, the rutting plate can be simplified as finite thickness elastic layer. Based on the analysis of finite thickness elastic layer subject to a circular surface pressure, the stress of the rutting plate can be approximated as
(8)$$\begin{array}{c} \sigma_{z}\left(t\right)=\alpha \sigma \left(t\right)\\ \sigma_{r}\left(t\right)=\beta \sigma \left(t\right)\\ \sigma_{\theta }\left(t\right)=\gamma \sigma \left(t\right) \end{array}$$
where α, β and γ can be calculated by boundary conditions, σ(t) describes the stress changes of the rutting plate above the PZT sensor. Substituting the Equations (Equation 7) and (Equation 8) into Equation (Equation 5), the correlation between output voltage and the wheel load σ(t) can be simplified as
(9)$$\frac{{\bar{\mu }}_{33}}{h_{p}m}\frac{dV}{dt}+\frac{V}{mRA_{p}}=-\frac{d\sigma (t)}{dt}$$
where $m=\frac{\sqrt{2}\;{\bar{d}}_{31}}{E}\left[\beta -\nu \left(\alpha +\gamma \right)\right]+\frac{{\bar{d}}_{33}}{E}\left[\alpha -\nu \left(\beta +\gamma \right)\right]$. The wheel load σ(t) can be obtained by integrating Equation (Equation 9) with respect to time *t*.

### 3.3. Identification Theory of MTS Test

The contact area between the rutting plate and load area of the MTS test is much larger than the wheel tracking test. Therefore, the Possion effect can be ignored and the thickness-wise electric displacement may be simplified as
(10)$$D_{z}={\bar{d}}_{33}\varepsilon_{z,p}+{\bar{\mu }}_{33}\frac{V}{h_{p}}$$
where ${\bar{d}}_{33}=d_{33}c_{33}$, ${\bar{\mu }}_{33}=\mu_{33}-d_{33}^{2}$. Correspondingly, the differential equations about voltage and strain can be simplified as
(11)$$\frac{dV}{dt}+\frac{Vh_{p}}{{\bar{\mu }}_{33}RA_{p}}=-\frac{h_{p}{\bar{d}}_{33}}{{\bar{\mu }}_{33}}\frac{d\varepsilon_{z,p}}{dt}$$

The load applied to rutting plate can be simplified as bowl-shaped distribution and along the radial direction can be read as
(12)$$p\left(r\right)=\left\{\begin{array}{cc} \eta p & r<a\\ 0 & r\geq a \end{array}\right.$$
where $p=\frac{1}{\pi a^{2}}\int_{0}^{a}2\pi rdr$, $\eta =\frac{3}{2}{\left(1-\frac{r^{2}}{a^{2}}\right)}^{0.5}$, *a* is the radius of loads. Then, the strain of PZT sensors can be calculated as
(13)$$\varepsilon_{z,p}=\varepsilon_{z,b}=\frac{\eta p(t)}{E}$$
where p(t) describes the changes of loads on the surface of rutting plate. Substituting Equation (Equation 13) into Equation (Equation 11), the correlation between output voltage and the load p(t) can be simplified as
(14)$$\frac{dV}{dt}+\frac{Vh_{p}}{{\bar{\mu }}_{33}RA_{p}}=-\frac{\eta h_{p}{\bar{d}}_{33}}{{\bar{\mu }}_{33}}\frac{dp(t)}{dt}$$

The load p(t) can be calculated by integrating Equation (Equation 14) with respect to time *t*.

## 4. Results and Discussion

The material properties of the PZT sensors used in our experiments are provided by the manufacturer. The material, geometrical, and circuit properties of PZT chips are $d_{31}=2.75\times 10^{-10}$ CN${}^{-1}$, $d_{33}=6.50\times 10^{-10}$ CN${}^{-1}$, $\mu_{33}=3.41\times 10^{-8}$ Fm${}^{-1}$, and its Young’s modulus is E = 65 GPa. To study the effect of location and embedded depth on the results, five PZT sensors were embedded in different locations of the rutting plate and each PZT sensor was placed at 4 different depths in the wheel test (10 mm, 20 mm, 30 mm, 40 mm). The results of the No.2 PZT sensor embedded in four different depths and four PZT sensors (No.1–4) embedded in the same depth 10 mm were chosen as an illustration. As the embedded depth increases, the voltage signals of the No.2 PZT sensor (Figure 3a) gradually decreases, indicating that the mechanical deformation of PZT gradually attenuates with the increasing depth. By inputting the voltage, the calculated zero-peak-zero stress (blue line in Figure 3c) agrees very well with the actual applied stress (red line in Figure 3c). Figure 3b describes the voltage signals of four PZT sensors (No.1–4) in the same depth 10 mm. As Table 2 showed, the identified load magnitude (Figure 3d) of No.1–4 PZT sensors are, respectively, 0.6955 MPa, 0.5389 MPa, 0.6203 MPa, 0.6955 MPa. The average identified load value of the four PZT sensors is 0.6375 MPa and its average error is 8.92%.

Compared with the wheel tracking test, the MTS test increased the loading frequency and magnitude. Besides, the contact area between the load and the rut board covered almost the entire rut board. Therefore, there was no stress attenuation along the depth direction. Figure 4 and Figure 5 describe the voltage signals and identified results of the No.1 PZT sensor with different loads (45 kN, 54 kN, 63 kN, 72 kN) at different frequencies (1 Hz, 5 Hz, 8 Hz) respectively. The calculated zero-peak-zero stress (blue line in Figure 5) increases with the loading frequency increasing. Table 3 describe the identified load results of the MTS test in detail, the maximum and minimum error are, respectively 18.70%, 0.88%. The average identified load results of three different frequencies are showed in Figure 6.

In moving load identification experiments, the trolley moved on the track at a certain speed with four different loads (26.5 kg, 46.5 kg, 66.5 kg, 86.5 kg). The stress applied to each wheel is considered approximately equal and the stresses with different loads are, respectively, 0.34 MPa, 0.59 MPa, 0.85 MPa, 1.1 MPa. The four different voltage peak signals (Figure 7a) correspond with the front wheel, rear wheel, rear wheel, and front wheel. It is obvious that the fourth voltage signal of each group is relatively weak. Therefore, the first three voltage signals of each group were selected to calculate. The average value of the identified results (Figure 8) under four different loads are, respectively, 0.42 Mpa, 0.62 MPa, 0.78 MPa, 0.90 MPa. The errors of identified results (Table 4) under four different loads are, respectively, 23.53%, 5.08%, 8.24%, 18.18%.

## 5. Conclusions

In summary, we proposed a theoretical electromechanical model that can correlate the output voltage signals generated by PZT sensors with the stress applied to it. A series of laboratory tests were performed to verify the accuracy of this model. Through the MTS test, we showed that the identified load magnitude increases as the loading frequency increases, which indicates that the loading frequency is a significant impact factor on the identified results. In the wheel tracking experiments, the maximum and minimum errors are, respectively 23.01%, 0.64%. In the MTS test, the maximum and minimum errors are, respectively 18.70%, 0.88%. In the moving load experiments, the maximum and minimum errors are, respectively 23.53%, 5.08%. According to the ASTM E1318 Standard Specification for Highway WIM Systems, almost all the relative errors of identified load results of three different experiments were at the range of permissible error ±20%. Both the identified results calculated by the model of cyclic load and moving load experiments exhibited a high accuracy, which suggests that this model is promising for application in improving the measurement accuracy of the WIM system. However, as this model ignores the dynamic effects of load, it might cause different identified results under the same load at different frequencies. Further investigation on the practical influence of load frequency in designing more suitable sensors for certain use scenarios remains an open topic for future study.

## Figures and Tables

**Figure 1 sensors-20-02366-f001:**
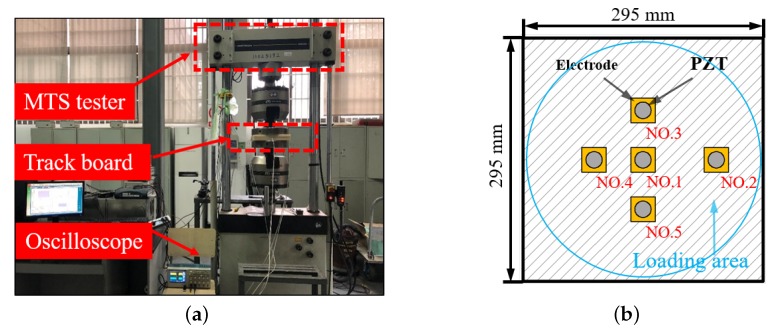
Mechanical Testing Simulation (MTS) test. (**a**) MTS test; (**b**) Loading area schematic diagram.

**Figure 2 sensors-20-02366-f002:**
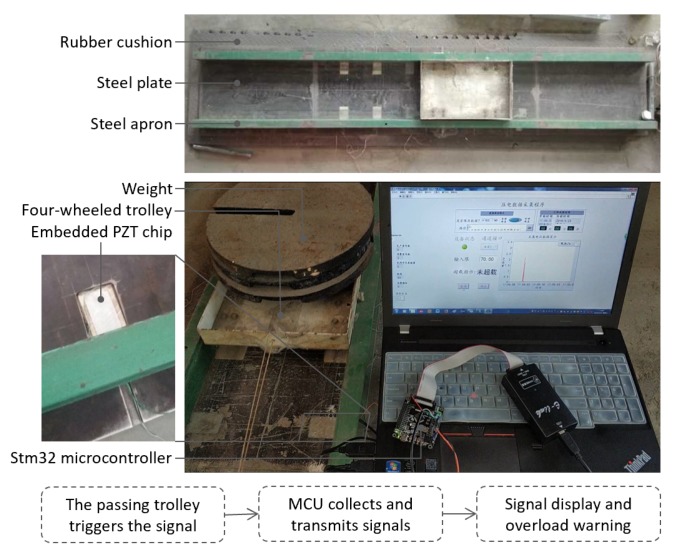
The moving load identified system.

**Figure 3 sensors-20-02366-f003:**
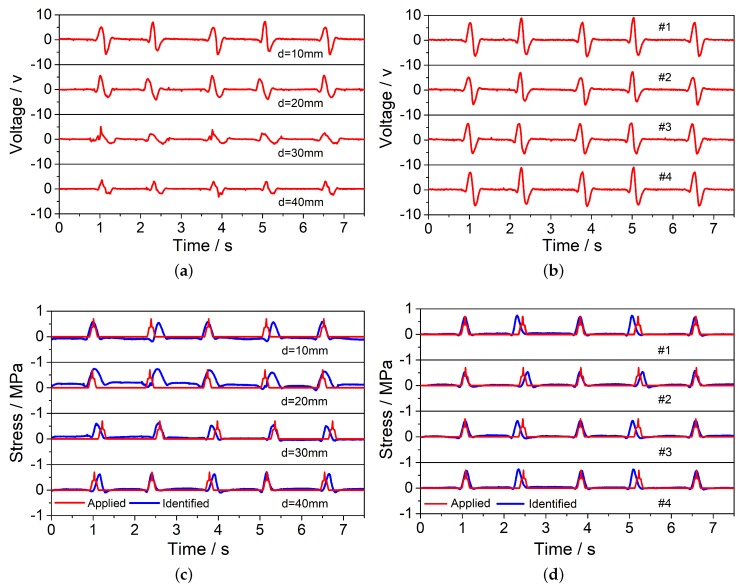
The voltage signals and identified results of the wheel tracking test. (**a**) Voltage signals of No.2 PZT chip at four different depths; (**b**) Voltage signals of No.1–4 PZT chips at same depth 10 mm; (**c**) Identified results of No.2 PZT chip at four different depths; (**d**) Identified results of No.1–4 PZT chips at same depth 10 mm.

**Figure 4 sensors-20-02366-f004:**
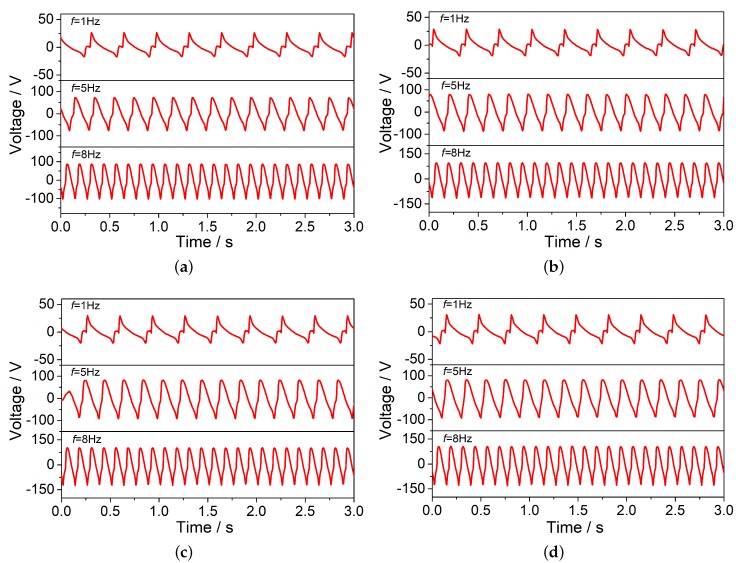
The voltage signals of the MTS test. (**a**) Voltage signals of No.1 PZT chip with three different loading frequencies under 45 kN load; (**b**) Voltage signals of No.1 PZT chip with three different loading frequencies under 54 kN load; (**c**) Voltage signals of No.1 PZT chip with three different loading frequencies under 63 kN load; (**d**) Voltage signals of No.1 PZT chip with three different loading frequencies under 72 kN load.

**Figure 5 sensors-20-02366-f005:**
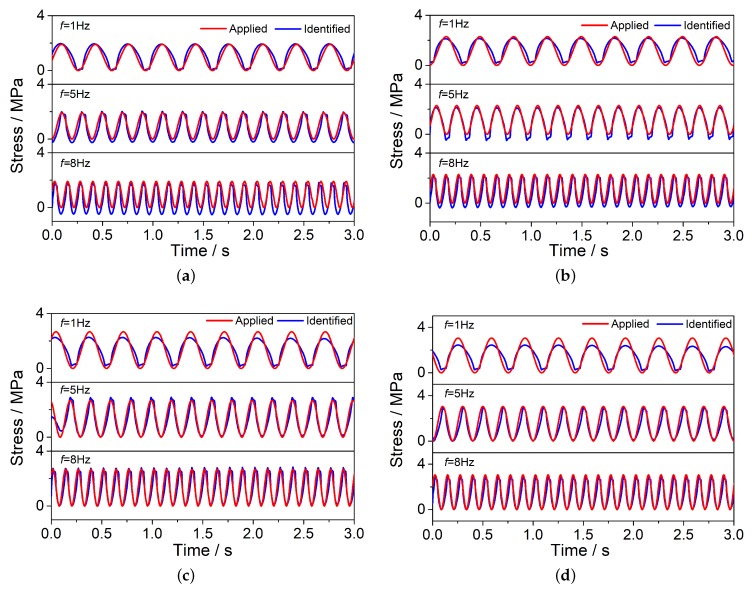
The Voltage signals and identified results of the MTS test. (**a**) Identified results of No.1 PZT chip with three different loading frequencies under 45 kN load; (**b**) Identified results of No.1 PZT chip with three different loading frequencies under 54 kN load; (**c**) Identified results of No.1 PZT chip with three different loading frequencies under 63 kN load; (**d**) Identified results of No.1 PZT chip with three different loading frequencies under 72 kN load.

**Figure 6 sensors-20-02366-f006:**
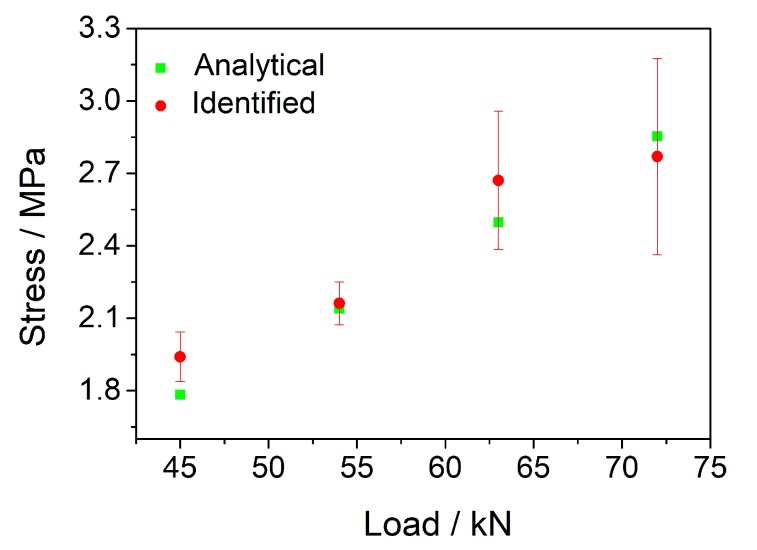
The average identified load results of three different frequencies.

**Figure 7 sensors-20-02366-f007:**
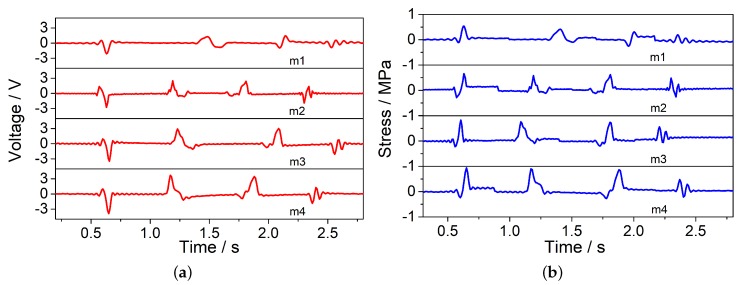
The results of moving loads experiments. (**a**) Voltage signals of PZT bimorph under four different loads (26.5 kg, 46.5 kg, 66.5 kg, 86.5 kg); (**b**) Identified results of PZT bimorph under four different loads (26.5 kg, 46.5 kg, 66.5 kg, 86.5 kg).

**Figure 8 sensors-20-02366-f008:**
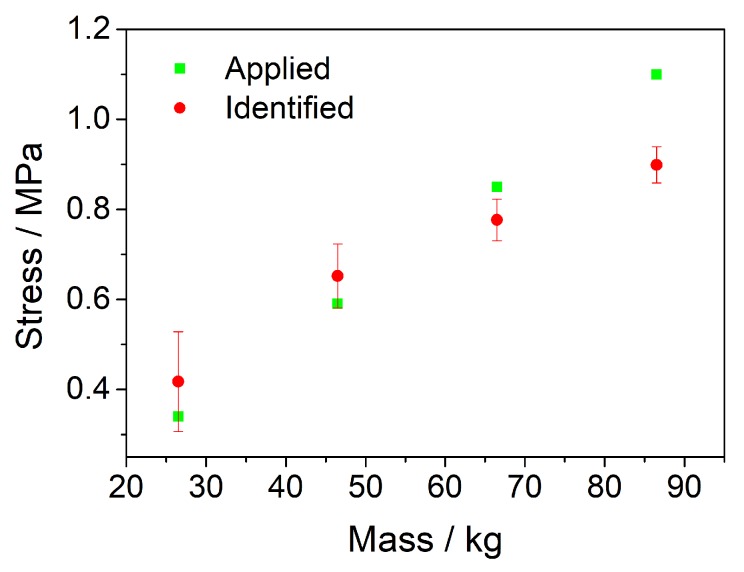
The average identified load results of moving load experiments.

**Table 1 sensors-20-02366-t001:** Sensor parameters.

Material	Area	Thickness	Piezoelectric Constant	Dielectric Constant
	$A_{P}$ (mm${}^{2}$)	$h_{p}$ (mm)	$d_{33}$ (pC/N)	$\mu_{33}$ (F/m)
PZT-chip	314	5	650	$3.41\times 10^{-8}$
PZT bimorph	192	0.2	600	$3.01\times 10^{-8}$

**Table 2 sensors-20-02366-t002:** The identified load results of No.1–4 Lead Zirconium Titanate (PZT) chips in the same depth (10 mm).

	NO.1	NO.2	NO.3	NO.4	Average
Applied Stress/MPa (A)	0.7	0.7	0.7	0.7	0.7
Identified Stress/MPa (I)	0.6955	0.5389	0.6203	0.6955	0.6376
Error $\frac{\left|I-A\right|}{A}$	0.0064	0.2301	0.1139	0.0064	0.0892

**Table 3 sensors-20-02366-t003:** The identified load results of No.1 PZT chip in the MTS test.

Load	Frequency	Identified Stress (I)	Theoretical Stress (T)	Error
(kN)	(Hz)	(MPa)	(MPa)	$\frac{\left|I-T\right|}{T}$
45	1	2.011	1.783	0.1274
54	1	2.101	2.140	0.0182
63	1	2.347	2.497	0.0600
72	1	2.320	2.854	0.1870
45	5	1.987	1.783	0.1143
54	5	2.121	2.140	0.0091
63	5	2.892	2.497	0.1581
72	5	3.110	2.854	0.0899
45	8	1.823	1.783	0.0222
54	8	2.263	2.140	0.0574
63	8	2.774	2.497	0.1111
72	8	2.879	2.854	0.0088

**Table 4 sensors-20-02366-t004:** The average identified load results of moving loads experiments.

	M1	M2	M3	M4
Mass/kg	26.5	46.5	66.5	88.5
Applied Stress/MPa (A)	0.34	0.59	0.85	1.1
Identified Stress/MPa (I)	0.42	0.60	0.70	0.90
Error $\frac{\left|I-A\right|}{I}$	0.2353	0.0508	0.0824	0.1818

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
