# Peer review of "Detection of Moving Load on Pavement Using Piezoelectric Sensors"

_sensors, 2020, doi:10.3390/s20082366_

Round 1
Reviewer 1 Report
The authors proposed a theoretical electromechanical model which can correlate signals generated by PZT sensors with the stress applied.
- The mathematical model is clear, however, the authors need to improve the paper showing a deeper bibliographic review of existing methods to justify the novelty.
- Piezo transducers are sensitivity to temperature variations. How does this model consider it? It is an important issue in real applications!
Reviewer 2 Report
Overview
The manuscript describes a prototype of a weigh-in-motion sensor system that can potentially be used for detection of overweight vehicles. The prototype is based on a simple PZT bimorph device embedded in a steel plate, that generates a voltage signal as it gets compressed by the load from the wheel of the vehicle.
The concept is interesting and useful from the engineering standpoint, but the paper needs to undergo a major revision prior to being accepted for publication. The authors must address the queries and suggestions listed below.
Reviewer’s remarks:
- The reviewer suggests to rephrase the title to more accurately represent the content presented by authors, so that it would read “Detection of Moving Loads on Pavement Using Piezoelectric Sensors”.
- Line 123- the authors report minimum errors achieved in the moving load and MTS tests. The reviewer believes that this information is selectively reported. The really important information in this case is the largest possible error. This must be reported, otherwise it brings up an issue of research and reporting ethics.
- Line 125- the authors state that “ Both the identified results calculated by the model of cyclic load and moving load experiments exhibited a high accuracy, which suggest that this model is of promising application in improving measurement accuracy for WIM system.“ The reviewer believes that it is necessary to state what is considered an acceptable amount of error in load calculation. Better yet, the authors shall quote acceptable accuracy specification for WIM systems either from the government agencies or industrial bodies. For example, ASTM E1318 - 09(2017) Standard Specification for Highway Weigh-In-Motion (WIM) Systems with User Requirements and Test Methods. With this information it will be possible to compare the errors reported for the new system with the accuracy requirements set forth by governing bodies. Please revise the conclusions section accordingly.
- Please report the identified load results presented in paragraphs on lines 112-117 in a table.
- It makes no sense why the authors describe results in Figures 4,5, and 6 before describing results in Figure 3. Please reorder figures.
- Please have the manuscript professionally edited for English language. There are too many issues with grammar, poor word choices, and general sentence flow.
Author Response
Sensors-762505
Dear Editor,
Thank you for giving us the opportunity to submit a revised draft of the manuscript entitled “Inversion of Moving Load on Pavement Using Piezoelectric Sensors” (Sensors-762505) for your further consideration. We appreciate the time and efforts that you and the reviewers dedicated to providing feedback on our manuscript and are grateful for the insightful comments on and valuable improvements to our paper. We have made modifications according to the reviewers’ comments in the revised manuscript. The point-by-point responses to the reviewers’ comments are listed below, and modifications to the manuscript have been made correspondingly.
Point 1: The reviewer suggests to rephrase the title to more accurately represent the content presented by authors, so that it would read “Detection of Moving Loads on Pavement Using Piezoelectric Sensors”.

Response 1: We thank the reviewer for this suggestion. Accordingly, we revised the title as “Detection of Moving Load on Pavement Using Piezoelectric Sensors”.
Point 2: Line 123- the authors report minimum errors achieved in the moving load and MTS tests. The reviewer believes that this information is selectively reported. The really important information in this case is the largest possible error. This must be reported, otherwise it brings up an issue of research and reporting ethics.
Response 2: Thank you for pointing this out. To show the identified load results clearly and comprehensively, we present the identified load results on following tables. Table.2 exhibits the identified load results of No.1-4 PZT chips in the same depth (10 mm).The maximum and minimum errors are respectively 23.01%, 0.64%.
In fact, five different loads (36kN, 45kN, 54kN, 63kN, 72kN) were performed in the MTS test and the average errors of three different frequencies were, respectively 9.08%, 13.54% and 6.18%. In the manuscript, we choose the results of the NO.1 PZT with four loads (45kN, 54kN, 63kN, 72kN) as an illustration. Table.3 shows the identified loads results in detail. The maximum and minimum errors are respectively 18.70%, 0.88%. Figure.4 shows the average identified load value of three different frequencies.Table.4 shows the identified load results of moving loads with four different mass. The maximum and minimum errors are respectively 23.53%, 5.08%. Accordingly, we revised the manuscript by adding these tables and figure.
Table.2 The identified load results of No.1-4 PZT chips in the same depth (10 mm)
|
|
NO.1 |
NO.2 |
NO.3 |
NO.4 |
Average |
|
|
Applied Stress/MPa (A) |
0.7 |
0.7 |
0.7 |
0.7 |
0.7 |
|
|
Identified Stress/MPa (I) |
0.6955 |
0.5389 |
0.6203 |
0.6955 |
0.6376 |
|
|
Error / |
0.0064 |
0.2301 |
0.1139 |
0.0064 |
|
Figure.3 The average identified load value of three different frequencies
Table.3 The identified load results of No.1 PZT chip in MTS test
|
Load
(kN) |
F
(HZ) |
Identified stress (I)
(MPa) |
Theoretical stress (T)
(MPa) |
Error |
|
|
45 |
1 |
2.011 2.101 2.347 2.320 1.987 2.121 2.892 3.110 1.823 2.263 2.774 2.879 |
1.783 2.140 2.497 2.854 1.783 2.140 2.497 2.854 1.783 2.140 2.497 2.854 |
0.1274 0.0182 0.0600 0.1870 0.1143 0.0091 0.1581 0.0899 0.0222 0.0574 0.1111 0.0088 |
|
|
54 |
1 |
||||
|
63 |
1 |
||||
|
72 |
1 |
||||
|
45 |
5 |
|
|||
|
54 |
5 |
||||
|
63 |
5 |
||||
|
72 |
5 |
||||
|
45 |
8 |
|
|||
|
54 |
8 |
||||
|
63 |
8 |
||||
|
72 |
8 |
Table 4 The identified load results of moving loads with four different mass
|
|
M1 |
M2 |
M3 |
M4 |
|
Mass / kg |
26.5 |
46.5 |
66.5 |
86.5 |
|
Applied Stress / MPa (A) |
0.34 |
0.59 |
0.85 |
1.1 |
|
Identified Stress / MPa (I) |
0.42 |
0.62 |
0.7 |
0.90 |
|
Error / |
0.2353 |
0.0508 |
0.0824 |
0.1818 |
Figure.4 The average identified load results of three different frequencies
Our modifications in the manuscript:
Line 147-153:
In the wheel tracking experiments, the maximum and minimum errors are, respectively 23.01%, 0.64%. In the MTS test, the maximum and minimum errors are, respectively 18.70%, 0.88%. In the moving load experiments, the maximum and minimum errors are, respectively 23.53%, 5.08%. According to the ASTM E1318 - 09(2017) Standard Specification for Highway WIM Systems, almost all the relative errors of identified load results of three different experiments were at the range of permissible error ± 20%. Both the identified results calculated by the model of cyclic load and moving load experiments exhibited a high accuracy, which suggests that this model is of promising application in improving measurement accuracy for the WIM system
Point 3: Line 125- the authors state that “ Both the identified results calculated by the model of cyclic load and moving load experiments exhibited a high accuracy, which suggest that this model is of promising application in improving measurement accuracy for WIM system.“ The reviewer believes that it is necessary to state what is considered an acceptable amount of error in load calculation. Better yet, the authors shall quote acceptable accuracy specification for WIM systems either from the government agencies or industrial bodies. For example, ASTM E1318 - 09(2017) Standard Specification for Highway Weigh-In-Motion (WIM) Systems with User Requirements and Test Methods. With this information it will be possible to compare the errors reported for the new system with the accuracy requirements set forth by governing bodies. Please revise the conclusions section accordingly.
Response 3: Thanks for your suggestions. We checked the ASTM E1318 - 09(2017) Standard Specification for Highway Weigh-In-Motion (WIM) Systems and revised the manuscript.
Our modifications in the manuscript:
Line 144-149:
In wheel tracking experiments, the maximum and minimum errors are, respectively 23.01%, 0.64%. In the MTS test, the maximum and minimum errors are, respectively 18.70%, 0.88%. In moving load experiments, the maximum and minimum errors are, respectively 23.53%, 5.08%. According to the ASTM E1318 - 09(2017) Standard Specification for Highway Weigh-In-Motion (WIM) Systems, almost all the errors of identified load results of three different experiments were at the permissible range of ±20%.
Point 4: Please report the identified load results presented in paragraphs on lines 112-117 in a table.
Response 4: We thank the reviewer for this comments. We present the identified load results on table.6 and add it in our manuscript.
Table 4 The identified load results of moving loads with four different mass
|
|
M1 |
M2 |
M3 |
M4 |
|
Mass / kg |
26.5 |
46.5 |
66.5 |
86.5 |
|
Applied Stress/MPa (A) |
0.34 |
0.59 |
0.85 |
1.1 |
|
Identified Stress/MPa (I) |
0.42 |
0.62 |
0.7 |
0.90 |
|
Error / |
0.2353 |
0.0508 |
0.0824 |
0.1818 |
Point 5: It makes no sense why the authors describe results in Figures 4, 5, and 6 before describing results in Figure 3. Please reorder figures.
Response 5: Thank you for pointing this out. We have revised it in our manuscript. The new numbers of Figure 3 is Figure 8.
Point 6: Please have the manuscript professionally edited for English language. There are too many issues with grammar, poor word choices, and general sentence flow.
Response 6: Thank you for pointing this out. We have revised this manuscript carefully.

Reviewer 3 Report
The manuscript presents experimental studies on inversion of moving Load on pavement using PZT sensors. The manuscript offers innovative results. Hence, the paper is recommended for publication after the following questions were addressed.
- In order to clarify the importance of the study, more literature review regarding PZTs applications in load monitoring and structural health monitoring should be added to the introduction section. Following papers can be reviewed in this regard:
- Aubryn Cooperman, Marcias Martinez,”Load monitoring for active control of wind turbines”, Renewable and Sustainable Energy Reviews, Volume 41,2015,Pages 189-201,
- Xin Wu, Dian Jiao, Lan You, “Nonintrusive on-site load-monitoring method with self adaption”, International Journal of Electrical Power & Energy Systems, Volume 119, 2020,
- Tashakori, A. Baghalian, M. Unal et al., “Contact and noncontact approaches in load monitoring applications using surface response to excitation method,” Measurement, vol. 89, pp. 197–203, 2016.
- Tashakori, A. Baghalian, V.Y. Senyurek, M. Unal, D. McDaniel, I.N. Tansel, Implementation of heterodyning effect for monitoring the health of adhesively bonded and fastened composite joints, Appl. Ocean Res. 72 (Mar. 2018) 51–59.
- Shima Taheri, “A review on five key sensors for monitoring of concrete structures”, Construction and Building Materials, Volume 204, 2019, Pages 492-509,
....
- In line 108 and figure 5 and 6, the authors used the ‘Kn’ unit for loads which I believe it should be change to ‘kN’.
- Authors mentioned the error identification in lines 110 and 117 which are not following any sort of pattern and need to be discussed.
- There are some grammatical and spelling mistakes like:
Line 5: applied to
Line 17: traffic flow has been
Line 18: static methods no longer meet
Line 21: time-saving
Line 24: main factors
Line 26: conducted
…
Which need to be addressed.
Round 2
Reviewer 2 Report
The reviewer is grateful for improvements made to the manuscript by the authors. In this form the manuscript is suitable for publication.